# Human Papilloma Virus (HPV) Detection in Oral Rinse vs. Oral Sponge: A Preliminary Accuracy Report in Oral Cancer Patients

**DOI:** 10.3390/cancers16193256

**Published:** 2024-09-24

**Authors:** Vera Panzarella, Michela Buttà, Fortunato Buttacavoli, Giuseppina Capra, Alberto Firenze, Nicola Serra, Giovanna Giuliana, Giuseppe Pizzo, Giuseppina Campisi, Rodolfo Mauceri

**Affiliations:** 1Department of Precision Medicine in Medical, Surgical and Critical Care (Me.Pre.C.C.), University of Palermo, 90127 Palermo, Italy; vera.panzarella@unipa.it (V.P.); giovanna.giuliana@unipa.it (G.G.); giuseppe.pizzo@unipa.it (G.P.); rodolfo.mauceri@unipa.it (R.M.); 2Center for Sustainability and Ecological Transition (CSTE), University of Palermo, 90133 Palermo, Italy; alberto.firenze@unipa.it; 3Department of Health Promotion, Mother and Childcare, Internal Medicine and Medical Specialties (ProMISE), University of Palermo, 90127 Palermo, Italy; michela.butta@unipa.it (M.B.); giuseppina.capra@unipa.it (G.C.); 4Unit of Oral Medicine and Dentistry for Fragile Patients, Department of Rehabilitation, Fragility, and Continuity of Care, University Hospital “Policlinico Paolo Giaccone” in Palermo, 90127 Palermo, Italy; giuseppina.campisi@policlinico.pa.it; 5Department of Neuroscience, Reproductive Sciences and Dentistry, Audiology Section University Federico II of Naples, 80131 Napoli, Italy; nicola.serra@unina.it

**Keywords:** human papillomavirus, oral squamous cell carcinoma, polymerase chain reaction, salivary diagnostics, human papillomavirus DNA test, HPV DNA

## Abstract

**Simple Summary:**

This research explores non-invasive methods for detecting Human Papillomavirus (HPV) in patients with oral squamous cell carcinoma (OSCC). The study compares two self-collection techniques, oral rinse vs. salivary sponge, to determine their effectiveness in identifying HPV DNA. Results from 26 patients indicated that the salivary sponge method was more accurate and sensitive compared to the oral rinse. This finding suggests that a salivary sponge may be a superior option, especially for patients with functional limitations caused by OSCC. Our research could lead to improved non-invasive diagnostic tools for HPV in clinical settings.

**Abstract:**

Background/Objectives: Human Papillomavirus (HPV) is a significant etiological factor in the development of oropharyngeal carcinogenesis. The detection of HPV in oral squamous cell carcinoma (OSCC) could be also crucial for diagnosis, prognosis, and treatment planning. This study compares the efficacy and accuracy of two non-invasive sampling methods, oral rinse, and oral sponge, in detecting HPV DNA in patients with OSCC. Methods: Twenty-six patients with histologically confirmed OSCCs were recruited (M/F = 15/11; mean age 68.6). From each patient, two self-collected oral specimens, in the form of an oral rinse and a salivary sponge (i.e., LolliSponge), were collected, and subsequently processed, utilizing INNO-LiPA HPV Genotyping Extra II for HPV DNA detection; Results: Oral sponge detection showed high specificity (100%), sensitivity (85.7%), and accuracy (96.2%) compared to the oral rinse sampling, also demonstrating an area AUC for its diagnostic performance significantly greater than 0.5 (0.93 vs. 0.5, *p* < 0.0001). Conclusions: This study supports that oral sponge sampling offers valuable non-invasive alternatives for HPV detection in patients with OSCC, with the potentiality to facilitate saliva sampling in patients that may exhibit functional deficit due to OSCC. Further research is recommended to validate these findings in larger cohorts and to explore the integration of these methods into routine clinical practice for the management of HPV-related OSCC.

## 1. Introduction

Oral squamous cell carcinoma (OSCC) is one of the most common types of head and neck cancers, accounting for over 90% of malignancies in the oral cavity [1]. Traditionally, OSCC has been strongly associated with risk factors such as tobacco use, alcohol consumption, and betel nut chewing [2]. However, over the past few decades, a growing body of epidemiological evidence has highlighted the significant role of Human Papillomavirus (HPV) in the etiology of head-neck SCC, particularly in the oropharyngeal district [3,4,5]. 

HPV is the most common sexually transmitted pathogen worldwide, with an estimated 80% of sexually active people contracting the infection at least once in their lifetime [6]. High-risk HPVs (hrHPV) are classically involved in the genesis of anogenital malignancies (i.e., cervical cancer) [7,8], however, the recent major changes witnessed in sexual behaviors have made oral sexual practices the most likely way of exposing the oral cavity to HPV infection and HPV-related carcinogenesis phenomena [9,10,11,12,13,14,15,16]. 

There has been a marked increase in the incidence of HPV-positive oropharyngeal squamous cell carcinoma (OPSCC), particularly in younger and non-smoker populations, data that contrasts with the traditional demographic of oropharyngeal cancer patients who are typically older and have a history of significant tobacco and alcohol use [17,18,19,20]. This shift suggests distinct biological behaviors and risk profiles between HPV-positive and HPV-negative oropharyngeal carcinogenesis [21,22]. 

There are no FDA-approved tests to detect HPV DNA or RNA in saliva; however, salivary rinse samples are usually used in research settings to assess oral HPV infection among both cancer patients and healthy people [23,24,25,26].

Oral rinse for HPV-DNA detection offers significant advantages, including non-invasiveness, ease of repeated testing, and cost-effectiveness. Particularly for patients with suspected/confirmed OSCC or oral potentially malignant disorders (OPMD), who may already be experiencing discomfort due to these oral suspected lesions or procedures for their diagnosis, oral rinse collection is particularly well accepted due to ease of use and less invasive nature. Moreover, oral rinse samples cells from the entire oral cavity, may increase the likelihood of detecting HPV if it is present, which could be important for identifying widespread or non-localized infections [27].

Challenges such as variable sample quality, potential contamination, and the need for advanced laboratory processing must be addressed to ensure accurate and reliable results. Variable sample quality could be attributed to inconsistent collection methods and the dilution effect of the liquid used for rinsing. Improper sample collection may vary depending on patient adherence to instructions and thoroughness during the rinsing process; and the dilution of saliva during rinsing can reduce the concentration of HPV DNA, potentially leading to false negatives, especially in cases of low viral load. Moreover, oral rinse samples can be contaminated by extraneous DNA from the environment, the container, or the patient’s hands, which may result in false positives [28]. The proximity between the oral cavity and oropharynx also exposes a risk of altered oral HPV detection, given the lack of site-specificity of sampling and the different susceptibility to infection known between the two districts. This is very important for appreciating the role of HPV in strictly oral carcinogenesis [14]. 

In this context, ensuring standardized collection protocols and performing other saliva sampling methods that can enhance diagnostic accuracy, especially for certain HPV-related cancers, such as OSCC, should be supposed. 

The LolliSponge (Copan Italia S.p.A., Brescia, Italy) is a novel device designed for saliva collection, commonly used for self-salivary sampling in active COVID-19 surveillance programs, known for its greater ease of use compared to oral rinse. The useful features of this saliva-direct device are that it is non-invasive, less expensive, and validated for use with reagents and instruments from multiple vendors and for several viral detections [29]. Its extreme versatility is associated with the greater adequacy of the sampling, due to the presence of a sponge which guarantees the scraping of the mucosa. This could translate into a more robust sample with reduced collection time and greater acceptance by the oncological patient, often with oral function already compromised by the cancer.

The main aim of this pilot study is to report the preliminary performance results of the LolliSponge sampling, compared a standard technique such as the oral rinse, in identifying oral HPV status, in a cohort of patients with strictly OSCC. 

## 2. Materials and Methods

### 2.1. Preliminary Accuracy Report

The study protocol adhered to the ethical principles outlined in the 1964 Declaration of Helsinki and its subsequent amendments or equivalent ethical standards. It was approved by the institutional review board of the University Hospital “Policlinico Paolo Giaccone” in Palermo, Italy (approval numbers #03/2013 and #04/2024). Prior to sample collection, all participants provided written informed consent.

### 2.2. Entry Criteria

Patient recruitment commenced on 1 January 2024 and concluded in June 2024. Participants were enrolled from the Oral Medicine Unit at the University Hospital “Policlinico Paolo Giaccone” in Palermo, Italy. 

The inclusion criteria were as follows:(i)Age 18 years or older;(ii)Ability to provide informed consent;(iii)Presence of suspected OSCC strictly within the oral cavity, categorized according to the 2024 NIH/SEER ICD-0-3.2 topographical classification codes;(iv)No prior cancer diagnosis or treatment in the head and neck regions.

### 2.3. Data Collection and Clinical Examination

Patients suspected of having OSCC were interviewed using a structured questionnaire to collect socio-demographic and medical history data. Regarding smoking and alcohol use, participants were categorized as never, current, or former smokers (those who had quit at least one year before the study) [2]. Alcohol consumption was classified as non-drinkers, moderate drinkers (fewer than 16 units per week), or heavy drinkers (16 or more units per week).

Lesions were categorized based on the 2024 NIH/SEER ICD-0-3.2 topographical codes used in the eligibility criteria [14], and grouped into the following categories:-Mobile tongue (including ventral/lateral tongue) (C020-C021-C022-C023, C028, C029);-Gum (including upper/lower gum and retromolar area) (C030, C031, C039, C062)-Hard palate (C050);-Buccal mucosa (C060, C061);-Floor of the mouth (C040, C041, C048-C049).

Local mechanical risk factors such as sharp cusps or incongruent prostheses were recorded, and details of overlapping lesion sites/codes were noted.

### 2.4. Saliva Samples Collection for DNA Extraction and HPV DNA Detection

Each recruited patient provided two collected oral specimens, in the form of an oral rinse and a salivary sponge, both sent to the Microbiology and Virology Unit of the University Hospital “Policlinico Paolo Giaccone” in Palermo for HPV detection. Each patient was instructed to abstain from food, drink, and oral hygiene products for at least 1 h before sampling. 

The oral rinse was performed by rinsing the oral cavity with 10 mL of Original Mint Scope mouthwash (Procter & Gamble, Cincinnati, OH, USA), which is considered the most suitable buccal cell collection medium for obtaining DNA for clinical and research applications [30]. Each sample was collected into a sterile 50 mL Falcon tube and centrifuged at 1600 rpm for 10′ to isolate the cellular component. The pellet was then resuspended in 1 to 4 mL of phosphate-buffered saline (PBS) and centrifuged again at 13,000 rpm for 5 min. After removing the supernatant, samples were stored at −20 °C or processed immediately. 

The second sampling consisted of collecting saliva through the absorbent sponge of the LolliSponge device (Copan Italia S.p.A., Brescia, Italy). The sample was then centrifuged at 1800 rpm for 1 min to extract the saliva from the sponge and stored at −20 °C or processed immediately. 

PBS resuspended oral pellet (200 uL each) and LolliSponge collected saliva underwent DNA extraction using the QIAamp Mini Kit (Qiagen, Hilden, Germany), following the manufacturer’s protocol. HPV-DNA was then detected by INNO-LiPA HPV Genotyping Extra II (Fujirebio, Tokyo, Japan), a reverse hybridization assay which identifies 20 hrHPV genotypes (HPV16, HPV18, HPV31, HPV33, HPV35, HPV39, HPV45, HPV51, HPV52, HPV56, HPV58, HPV59, HPV67, HPV68, HPV26, HPV53, HPV66, HPV70, HPV73, HPV82) and 12 lrHPV genotypes (HPV6, HPV11, HPV40, HPV42, HPV43, HPV44, HPV54, HPV61, HPV62, HPV81, HPV83, HPV89). 

Samples that were positive but could not be genotyped using the primary diagnostic kit underwent a nested PCR assay for enhanced sensitivity. This involved initial amplification with the PGMY09/11 primers, followed by a secondary amplification using GP05+/GP06+ primers, as described in previous studies [31]. Genotypes were determined through Sanger sequencing of the PCR products, followed by sequence alignment using the Basic Local Alignment Search Tool (BLAST).

The entire HPV sampling and detecting protocols, by the two oral devices, are described in Table 1. All saliva sampling using the LolliSponge device was conducted first, followed by the oral rinse with Scope mouthwash, immediately before the diagnostic biopsy for OSCC. This sequence was followed to minimize any potential interference between the two sampling methods, with all procedures taking place on the same day during patient preparation.

### 2.5. Tissue Sample for Histological Examination 

All patients underwent incisional biopsies of selected areas of suspected carcinoma, performed using a scalpel punch under local anesthesia. Biopsy samples were fixed in formalin and processed in the Pathology Laboratory of the University Hospital “Policlinico Paolo Giaccone” in Palermo. Formalin-fixed, paraffin-embedded (FFPE) tissue sections (5 µm thick) were stained with hematoxylin and eosin for histopathological examination to confirm OSCC diagnosis. Carcinomas were graded according to the WHO classification, and only OSCC coded as 807*/* by the ICD-0-3 SEER site/histology validation list were included.

### 2.6. Statistical Analysis 

Categorical variables were presented as counts and percentages, while continuous variables were expressed as mean ± standard deviation (SD) unless otherwise indicated. The diagnostic performance of the LolliSponge device for HPV-positive patients was assessed using sensitivity, specificity, and the Receiver Operating Characteristic (ROC) curve, which plots the true positive rate against the false positive rate across various cut-off points. The area under the ROC curve (AUC) was calculated with standard error and 95% confidence intervals. Comparisons between two AUCs were performed using the z-test. Statistical significance was set at a *p*-value (*p*) < 0.05. All analyses were conducted using MATLAB analytical toolbox version 2008 (MathWorks, Natick, MA, USA) for Windows at 32 bits.

## 3. Results

Twenty-six patients with OSCC histological diagnoses were definitively recruited. A majority of the patients were males (15/26, 57.7%), with a mean age of 68.6 ± 11.7 years (range 44–89 years). In Table 2, demographic/clinical and tobacco/alcohol data are reported. 

Regarding oral lesions sites, the tongue was the most affected site (15/26; 57.7%), followed by the gums (10/26; 38.7%), and the buccal mucosa (4/26; 15.4%). Some OSCC cases involved multiple anatomical sites simultaneously. 

Up to a quarter of the lesions were HPV-positive (7/26, 26.9%).

Regarding the seven HPV+ lesions, most of the lesions occurred on the gum (4/7), in detail, two on the mandibular gingiva (C031), and two on the retromolar trigone area (C062). The remaining three lesions were located on the tongue (3/7), two on the border of the tongue (C021), and one on the anterior 2/3 of the tongue (C023) (Table 2).

As presented in Table 3, the oral rinse with Scope Oral Rinse and LolliSponge investigations highlighted the presence of different HPV genotypes among the HPV-positive OSCCs. Specifically, the investigation showed the presence of the same HPV genotype in the HPV-positive OSCCs, with only two exceptions. On one OSCC (#5) on the anterior 2/3 of the tongue (C023), the Scope Oral Rinse revealed the presence of genotype 38, while the LolliSponge showed the presence of genotype 17, even if the latter was 38-related. One OSCC (#12), on the border of the tongue (C021), was associated with the 66 HPV genotype by the Scope Oral Rinse, while the LolliSponge did not reveal any HPV in the sample.

Moreover, in two cases (#13, #19), by both oral samples tested, beta-HPV genotypes were identified, and only in one case, even if the HPV control generic band resulted positive by INNO-LiPA, genotyping it was not possible due to the low amount of amplified viral DNA. 

Other risk factors, such as smoking status, alcohol consumption, and local trauma were detected in a minimum proportion of overall OSCC cases (11.5, 3.8%, and 7.7%, respectively). In HPV-positive OSCCs, no local trauma was identified, and only in one case of multiple HPV detection (#7), current smoking and moderate alcohol consumption were registered. 

Table 4 presents the performance of the LolliSponge diagnostic to individualize patients HPV positive, through statistical indices such as sensitivity, specificity, and accuracy, considered as the gold standard of the Scope Oral Rinse diagnostic. LolliSponge detection shows high specificity (100%) sensitivity (85.7%), and accuracy (96.2%) considering oral rinse diagnostic results. 

Figure 1 represents a rose-plot graph, where we illustrated the percentages of true negative (100%), true positive (85.7%), false negative (14.3%), and false positive (0.0%). For the rose-plot, we used the area of segments of the circle to convey amounts, where the angle is constant i.e., divided 360 by the number of parameters considered thus, the square root of the radius is proportional to percentages. 

To obtain a complete sensitivity/specificity report, the Operating Characteristic (ROC) curve analysis was applied. Particularly, the graph reported the area under the ROC curve (AUC) and compared to the area under the red line equal to the value 0.5. Figure 2 presents the area AUC associated with the LolliSponge diagnostic performance which was significantly greater than 0.5 (0.93 vs. 0.5, *p* < 0.0001). 

## 4. Discussion

The prevalence of HPV in OSCC varies widely across the world. Globally, it is estimated that approximately 25–30% of OSCC cases are HPV-positive, though this can range from as low as 6% in some countries to as high as 60% in others [5,32,33,34].

This discrepancy is attributed to the variability of oral HPV sampling and detection methods, as well as the lack of detail of the sites of lesions studied, not strictly including just the oral region but bordering the oropharyngeal region, which is known to be more susceptible to HPV infection [14]. Consequently, the use of HPV sampling techniques must be as site-specific as possible, such as fresh tissue of suspected lesion from biopsy, which would be preferred in research settings. Nevertheless, this procedure involves high professional experience and collaboration and is difficult to perform in conventional clinical practice, where more accessible approaches with the same diagnostic accuracy are required [26,35].

Non-invasive methods for collecting buccal epithelial cells include cotton swabs, cytology brushes, and oral rinse/mouthwashes. Oral rinse is a valuable sampling tool for detecting oral HPV due to its non-invasive nature, ease of use, and cost-effectiveness. It is widely adopted in clinical and research settings to investigate oral HPV status, both as a tool for screening in healthy people and as an adjunct test in patients with OSCC and potentially malignant oral lesions, to investigate the possible role of HPV in etiological prognosis and recurrence process of oral cancer [23,27,36]. 

However, it does have limitations regarding sample quality and sensitivity. Oral rinse samples, independent from carrier fluids used, might contain lower viral loads when compared to tissue biopsies, potentially leading to false negatives in individuals with low-level infections. Moreover, oral rinses might not effectively capture HPV infections localized in specific areas of the oral cavity, limiting their site-diagnostic accuracy [37,38]. Furthermore, an aspect that is not well valued is the potential difficulty of making a proper oral rinse according to standard protocol, by patients with muscle-function joint impairment by the presence of cancer (e.g., OSCC of the tongue or floor of the mouth).

The Copan LolliSponge is a novel saliva specimen collection system that can be particularly useful for diagnostic purposes, including DNA viral detection. It has been validated by employing a molecular test for the search for SARS-CoV-2 confirming it as an extremely robust collection sample (https://www.copangroup.com/product-ranges/lollisponge/, accessed on 20 August 2024).

This device simplifies the process by allowing individuals to collect their saliva samples using a sponge on a stick, which is held in the mouth for a few minutes. This method minimizes contamination risks and does not require rinse/gargle or spitting, making it suitable for self-and/or supported collection, without specific professional assistance. Moreover, the sample site specificity in HPV detection could be greatly improved by directing the sponge specifically on the lesion.

In our study, we compare the LolliSponge device with the oral rinse procedure using the most suitable medium for buccal cells/DNA collection tested in clinical and research testing (Scope Mouthwash) [30] to detect HPV status in a cohort of 26 patients with strictly oral OSCC. To the best of our knowledge, it is the first time that this device is proposed in the literature for oral HPV sampling and detection. 

The overall practical advantages vs. disadvantages detected by the two salivary sample devices are detailed in Table 5. 

Regarding the performance of oral accuracy HPV diagnostic, between the two salivary samples, our results show a high overlapping diagnostic accuracy of LolliSponge compared to Scope Oral Rinse scores. 

In one case of OSCC of the gum (#7), both the Scope Oral Rinse and LolliSponge demonstrated equal suitability to identify all hrHPV genotypes of multiple infections, likely supported by the presence of other risk factors (i.e., smoking and alcohol habits) and/or the susceptibility of the site to plaque-related inflammatory (i.e., gingivitis and/or periodontitis), supporting a hypothesis of a potential complex interplay between local periodontal diseases and oral HPV infections [39].

Only in one case the LolliSponge device was not able to confirm the same HPV positivity as detected by oral rinse. We believe that this discrepancy may be attributable to the full mouth sampling done with Scope Oral Rinse, possibly detecting the presence of HPV in other location than the OSCC site, where instead the LolliSponge had not identified any HPV-DNA. 

To endorse this hypothesis, it would be useful to validate the HPV detection results by these two salivary diagnostics methods with those arising from an adjunctive site-specific sample (i.e., fresh tissue fragment lesion), as done previously [14]. This will be the future aim of our research group, together with the validation of salivary diagnostics methods’ accuracy in a larger sample.

This study has some limitations and potential biases. First, the sample size is relatively small, though it is considered appropriate given the preliminary nature of the study. For the same reason, other potential confounders, such as metastatic cancer status and disease stage, have not been analyzed at this stage of the investigation. Secondly, while non-invasive sampling methods such as oral rinse and sponge are practical, they may result in lower viral load detection when compared to tissue biopsies, potentially leading to false negatives. Additionally, although the LolliSponge device showed promising results, its use for HPV detection in oral cavity lesions has not yet been extensively validated in larger, more diverse populations. Finally, operator variability in sample collection and handling could introduce inconsistencies. Addressing these limitations in future research would enhance the reliability and applicability of the findings.

## 5. Conclusions

In conclusion, in this preliminary study, the LolliSponge diagnostic device, used for the first time to individualize HPV status in OSCC patients, showed excellent performance compared to the oral rinse gold standard. This potential, to be validated in the future, could open new horizons for the non-invasive study of HPV prevalence strictly in the oral cavity.

## Figures and Tables

**Figure 1 cancers-16-03256-f001:**
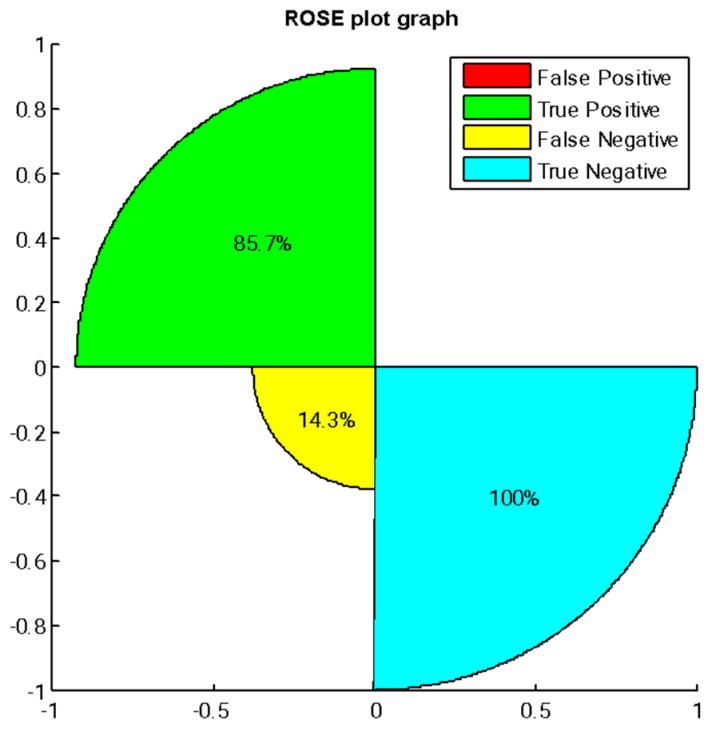
Rose-plot graph about the percentages of correct and incorrect cases diagnosed by LolliSponge diagnostic.

**Figure 2 cancers-16-03256-f002:**
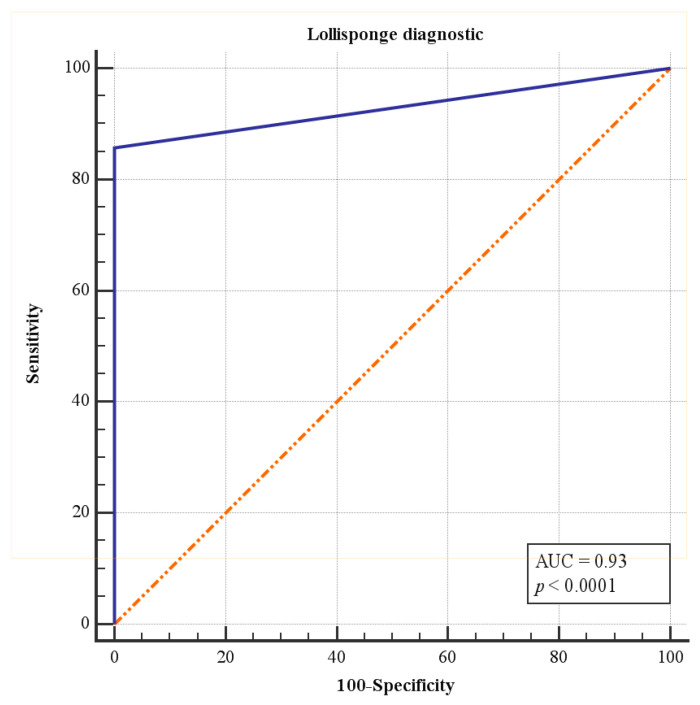
ROC curve analysis for the LolliSponge diagnostic performance for HPV positive patients. AUC under the red line is equal to 0.5.

**Table 1 cancers-16-03256-t001:** Oral HPV sampling and detecting protocols by Scope Oral Rinse and LolliSponge.

Step	Description	Procedures
**1**	Sample collection	Scope Oral Rinse: (i) dispense 10 mL of Original Mint Scope mouthwash into a sterile 50 mL Falcon tube; (ii) rinse orally with the mouthwash for 60 s, carefully reaching all parts of the mouth, avoiding gargling; then, spitting back into the Falcon tube. LolliSponge: (i) open the LolliSponge test tube holding the device by the cap and insert it in the mouth; (ii) gently move it around the mouth and scrape into suspected lesions, for 60 s, so that the sponges are well moistened; (iii) place the cap on the LolliSponge test tube, screw it on and securely close the test tube.
**2**	Sample process	Scope Oral Rinse: (i) centrifuge at 1600 rpm for 10′ to isolate the cellular component; (ii) resuspended the pellet in 1 to 4 mL of phosphate-buffered saline (PBS) and centrifuged again at 13,000 rpm for 5 min; (iii) after removing the supernatant, store the samples at −20 °C or process immediately. LolliSponge: (i) centrifuge at 1800 rpm for 10′ to isolate saliva form the absorbent sponge by centrifugating the device at 1800 rpm for 1 min; (ii) after removing the supernatant, store the samples at −20 °C or process immediately.
**3**	HPV-DNA extraction	Scope Oral Rinse and LolliSponge Use QIAamp Mini Kit (Qiagen, Hilden, Germany) to extract DNA following manufacturers’ protocol.
**4**	HPV detection and genotype	Scope Oral Rinse and LolliSponge Use INNO-LiPA HPV Genotyping Extra II (Fujirebio, Tokyo, Japan) and Basic Local Alignment Search Tool (BLAST) to identify HPV genotypes. Perform Nested PCR, followed by Sanger sequencing and Basic Local Alignment Search Tool (BLAST) alignment, for samples positive to Inno-lipa HPV controls, but not genotyping by the diagnostic kit.

**Table 2 cancers-16-03256-t002:** Information regarding age, sex, oral lesions site, local trauma, and smoking/alcohol consumption.

OSCC (Total No. 26)		% (No.)
**Gender**	Male	57.7% (15)
	Female	42.3% (11)
**Age Groups**	≤50	7.7% (2)
	51–60	19.2% (5)
	61–70	30.8% (8)
	>70	42.3% (11)
**Mean ± SD**	67.5 ± 11.7	
**Median (IQR)**	68.5 (60, 76)	
**Site**	Mobil tongue (C021, C022, C023)	57.7% (15) *
	Gum (C030, C031, C062)	38.7% (10) *
	Buccal mucosa (C060)	15.4% (4) *
	Hard palate (C050)	3.8% (1) *
	Floor of mouth (C041)	7.6% (2) *
**HPV status**	Positive	26.9% (7)
	Negative	73.1% (19)
**Local Trauma**	No trauma	92.3% (24)
	Presence of trauma	7.7% (2)
**Smoking Status**	Never	69.3% (18)
	Current	11.5% (3)
	Former	19.2% (5)
**Alcohol Consumption**	Non-drinker	96.2% (25)
	Moderate	3.8% (1)

* some OSCC presented at multiple anatomical sites simultaneously (see Table 2 for details).

**Table 3 cancers-16-03256-t003:** Comprehensive HPV diagnostic results by anatomical site of OSCC and risk factors from the oral rinse with Scope Oral Rinse and LolliSponge. The site codes are categorized using NIH/SEER ICD-0-3.2 topographical classification codes; ‘-‘ indicates a negative result.

No. Case	Sex	Site	Scope	LolliSponge	Local Trauma	Smoking Status	Alcohol Consumption
**#1**	F	Upper gums–Lower gums	-	-	No	Never	Non-drinker
**#2**	M	Buccal mucosa	-	-	No	Current	Non-drinker
**#3**	F	Buccal mucosa–Hard palate	-	-	No	Never	Non-drinker
**#4**	M	Buccal mucosa	-	-	No	Former	Non-drinker
**#5**	**M**	**Anterior 2/3 of tongue**	**38**	**17** (38 related)	**No**	**Former**	**Non-drinker**
**#6**	**M**	**Retromolar area**	**16**	**16**	**No**	**Never**	**Non-drinker**
**#7**	**M**	**Retromolar area**	**56 62 66 68**	**56 62 66 68**	**No**	**Current**	**Moderate**
**#8**	M	Ventral surface of tongue–Lateral floor of mouth	-	-	No	Never	Non-drinker
**#9**	F	Border of tongue	-	-	No	Never	Non-drinker
**#10**	F	Border of tongue	-	-	No	Never	Non-drinker
**#11**	F	Border of tongue	-	-	No	Never	Non-drinker
**#12**	**M**	**Border of tongue**	**66**	**-**	**No**	**Former**	**Non-drinker**
**#13**	**F**	**Border of tongue**	**beta-HPV**	**beta-HPV**	**No**	**Never**	**Non-drinker**
**#14**	M	Border of tongue	-	-	No	Former	Non-drinker
**#15**	M	Border of tongue	-	-	No	Former	Non-drinker
**#16**	M	Border of tongue– Ventral surface of tongue	-	-	Presence of Trauma	Never	Non-drinker
**#17**	F	Buccal mucosa	-	-	Presence of Trauma	Never	Non-drinker
**#18**	F	Lower gums–Lateral floor of mouth	-	-	No	Never	Non-drinker
**#19**	**F**	**Lower gums**	**120**	**120**	**No**	**Never**	**Non-drinker**
**#20**	M	Dorsal surface of tongue	-	-	No	Never	Non-drinker
**#21**	M	Border of tongue	-	-	No	Current	Non-drinker
**#22**	M	Ventral surface of tongue–Anterior 2/3 of tongue	-	-	No	Never	Non-drinker
**#23**	F	Upper gums	-	-	No	Never	Non-drinker
**#24**	**F**	**Lower gums**	**K1/K2**	**K1/K2**	**No**	**Never**	**Non-drinker**
**#25**	M	Lower gums	-	-	No	Never	Non-drinker
**#26**	M	Lower gums	-	-	No	Never	Non-drinker

**Table 4 cancers-16-03256-t004:** LolliSponge diagnostic test performance parameters about HPV-positive patients.

	Sensitivity (CI at 95%)	Specificity (CI at 95%)	Accuracy (CI at 95%)
LolliSponge	85.7% (65.5%, 96.7%)	100% (84%, 100%)	96.2% (78.4%, 100%)

**Table 5 cancers-16-03256-t005:** Advantages and disadvantages of Scope Oral Rinse and LolliSponge applications.

Application	Scope Oral Rinse		LolliSponge	
**Sample Collection**	Non-invasive, painless, easy to perform; requires patient adherence to instructions.	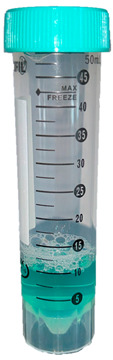	Non-invasive, painless, and less expensive; greater ease of use also without patient’s compliance, reducing sample collection time.	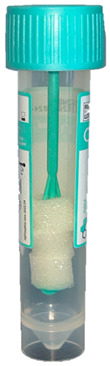
**Patient Acceptance**	Acceptance due to ease of use and less invasive nature.		Particularly beneficial for patients already experiencing oral discomfort.	
**Sample Adequacy**	Samples cells from the entire oral cavity; increases likelihood of detecting non-localized HPV infections.		Ensures mucosal scraping, potentially leading to more adequate samples.	
**Potential Contamination**	Risk of contamination from extraneous DNA, leading to false positives.		Reduced contamination risk compared to oral rinse.	

## Data Availability

Data available on request.

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
