# Peer review of "Human Papilloma Virus (HPV) Detection in Oral Rinse vs. Oral Sponge: A Preliminary Accuracy Report in Oral Cancer Patients"

_cancers, 2024, doi:10.3390/cancers16193256_

Round 1
Reviewer 1 Report
Comments and Suggestions for Authors
Dear Authors, Thank you for your submission.
Although HPV has been proven to be the etiologic agent for many oropharyngeal cancers, it may not be the cause of oral squamous cell cancer even if HPV is found in the oral cavity. Could the finding of HPV be considered a bystander effect? ie HPV could be present but not causing the cancer? Is there any literature that supports increased risk of oral squamous cancer in patients that harbor HPV DNA in the oral cavity?
Pathologic examination of HPV related oropharyngeal cancer has a characteristic non-keratinizing basaloid appearance - Did your oral cancer cases with positive HPV have a similar histologic appearance?
In Table 3, please also include the HPV types found in the actual biopsy specimens as a separate column; also it would be useful to have the actual site name ie "gums" rather than the code C030 in the table; for patients #1,#,8,16,18,22 where there were 2 codes - did these patients have 2 separate biopsies with cancer? Patient #19 ?120 is this the HPV subtype?; Patient #24 K1/K2 please explain this finding?
What was the timing between the biopsy and the patient sampling using oral rinse/Lollisponge? days? weeks?
Comments on the Quality of English LanguageThere are a few spelling errors
Author Response
Response to Reviewer 1’s Comments - Manuscript ID: cancers-3191603
Dear Reviewer 1,
Thank you for your thoughtful comments and questions regarding our manuscript titled "Human Papilloma Virus (HPV) Detection in Oral Rinse vs Oral Sponge: a Preliminary Accuracy Report in Oral Cancer Patients"(Manuscript ID: cancers-3191603). We appreciate the opportunity to address these points and provide additional clarification.
Comment 1:
Although HPV has been proven to be the etiologic agent for many oropharyngeal cancers, it may not be the cause of oral squamous cell cancer even if HPV is found in the oral cavity. Could the finding of HPV be considered a bystander effect? i.e., HPV could be present but not causing the cancer? Is there any literature that supports increased risk of oral squamous cancer in patients that harbor HPV DNA in the oral cavity?
- Response 1:
HPV is a well-established causative agent for oropharyngeal cancers, but its role in oral squamous cell carcinoma (OSCC) remains less clear. The prevalence of HPV in oral lesions varies widely, ranging from 0% to 85% across different studies, which may be influenced by the sampling techniques used and the specific sites investigated. In OSCC specifically (strictly excluding oropharyngeal sites), HPV prevalence is reported to range from 0% to 18% (https://doi.org/10.1007/978-3-030-32316-5_4). However, very few studies adhere to this strict distinction of anatomical sites and the standardization of HPV detection techniques (https://doi.org/10.3390/cancers13184595). As a result, HPV's direct role in OSCC carcinogenesis is not as well established as it is for oropharyngeal cancers. Further research is needed to clarify the exact relationship and to distinguish between potential bystander effects and causative roles. In this study, we compare the diagnostic accuracy of two non-invasive salivary sampling methods for HPV-DNA detection in strictly OSCC patients. The main aim is to report the preliminary performance results of an oral sponge sampling method (LolliSponge), compared to a standard technique, the oral rinse, in identifying oral HPV status in a cohort of OSCC patients. This is the first step in a larger project aimed at standardizing HPV identification techniques in the oral cavity. Our future goal will be to use this protocol to investigate the true etiological role of HPV in oral carcinogenesis, using targeted detection techniques (e.g., HPV methylation).
Comment 2:
Pathologic examination of HPV-related oropharyngeal cancer has a characteristic non-keratinizing basaloid appearance - Did your oral cancer cases with positive HPV have a similar histologic appearance?
- Response 2:
We appreciate your query regarding the histological characteristics of HPV-positive oral cancers. Nevertheless, in this first step of our investigation, the tissue samples were used only for histological confirmation of OSCC. No direct and/or indirect investigations of HPV on a histological specimen were carried out on this population because it is currently outside the main objectives of this preliminary study.
Comment 3:
In Table 3, please also include the HPV types found in the actual biopsy specimens as a separate column; also, it would be useful to have the actual site name i.e., "gums" rather than the code C030 in the table; for patients #1, #8, #16, #18, #22 where there were 2 codes - did these patients have 2 separate biopsies with cancer? Patient #19 ?120 is this the HPV subtype?; Patient #24 K1/K2 please explain this finding?
- Response 3:
Thank you for your insightful comments and suggestions regarding Table 3. Regarding the inclusion of HPV types from the biopsy specimens, we would like to clarify that our current study protocol does not include HPV typing in biopsy samples. Therefore, these results are not available for inclusion in the table currently. However, as per your suggestion, we have revised Table 3 to replace the site codes with specific site names for better clarity.
For patients with OSCC involving multiple sites, biopsies were performed at different locations when clinically necessary for diagnostic purposes. For patient #19, the code "120" corresponds to HPV 120, a known cutaneous HPV genotype from the betapapillomavirus 2 group. Additionally, the notation K1/K2 for patient #24 refers to a positive INNO-LiPA test result for which the specific genotype could not be determined. To briefly explain, the INNO-LiPA HPV Genotyping Extra II is a line probe assay based on reverse hybridization following PCR amplification using SPF10 primers. The results are interpreted by reading lines on a membrane strip, which includes: a conjugate control line (to ensure the procedure was correct), a human DNA line (to serve as a quality and extraction control), two lines indicating general HPV positivity, and 36 lines each corresponding to a specific HPV genotype. Sample #24 was positive for the two generic HPV lines but not for any of the specific genotype lines.
Comment 4:
What was the timing between the biopsy and the patient sampling using oral rinse/Lollisponge? Days? Weeks?
- Response 4:
All saliva sampling using the LolliSponge device was conducted first, followed by the oral rinse with Scope mouthwash, immediately before the biopsy for histological OSCC confirmation. This sequence was followed to minimize any potential interference between the two sampling methods, with all procedures taking place on the same day during patient preparation. We have included this information in the Material and Methods (lines 270-274) of the revised manuscript to provide a clearer understanding of the potential impact of this timing on our results.
Comment 5: There are a few spelling errors.
- Response 5: We appreciate your attention to the quality of the English language. We have thoroughly proofread the manuscript and corrected the spelling errors. We have also ensured that the language is clear and precise.
We made the necessary revisions and incorporated the additional information into the manuscript. We believe these changes address your concerns and enhance the clarity and completeness of our study. Please find the revised manuscript attached for your review.
Thank you once again for your valuable feedback.
Best regards,
Fortunato Buttacavoli
Corresponding Author
DDS, PhD Student, Postgraduate Specialist in Pediatric Dentistry
Department of Precision Medicine in Medical, Surgical and Critical Care (Me.Pre.C.C.)
University of Palermo, Via Liborio Giuffrè, 5 - 90127
fortunato.buttacavoli@unipa.it

Reviewer 2 Report
Comments and Suggestions for Authors
OI have provied my comments in the attached.
I am requesting modifications.
Review
Title: Human Papilloma Virus (HPV) detection in oral rinse vs oral sponge: a
preliminary accuracy report in oral cancer patients
Strengths:
The general premise for this study is an important aspect of cancer control which is the need for non-diagnostic methods to assess oral microenvironment using a non-invasive oral rinse with scope or Lollisponge. This approach facilitates development of rapid chairside detection and enhance communication between oral health professional and the patient, possible reducing time for early treatment if needed.
The presence of HPV within the oral cavity was used as an example of the efficacy of the device, Lollisponge. However, the choice of detection of HPV is a poor selection for this demonstration.
HPV as a reservoir can mediate oral mucosal immunity to reduce tumor immune surveillance but this is not addressed in this study.
This study in contrast simply used HPV subtype detection as an example of specificity, sensitivity and accuracy provided by the “Lollisponge.
Weaknesses:
The primary question is: What is the value of the non-invasive technique Lolli sponge to detect important factors that characterize early malignant events linked to OSCC?
There are important requirement for any detection system, if it is to be more than just a mechanical approach of non-invasive technique. The brush non-invasive approach is not novel (e.g., oral cytology brushes), but has been employed in repeated studies for many years. Specifically, there is a US patent to use oral brushing using any device that provides a brushing action such as a Lollisponge, but importantly the patent describes use of oral brushing to obtain a miRNA classifier identifying OSCC compared to non-OSCC cells.
In brief malignant event drivers cause molecular-genomic alterations in the host oral keratinocyte-squamous cell (stem or stem cell like) that results in faulty genetic repairs and regulations with loss of check point cell cycle control and homeostatic maintenance. In addition, there is a depressive immunosuppression that occurs with loss of tumor immune surveillance. I mention these distinctive characteristics, as the state of the art for any cancer control or screening methodology as referenced above needs to comply with this need for assessing these characteristics.
Unfortunately, the approach described focuses on HPV detection which does not fulfill the requirements noted above and is therefore a poor choice to demonstrate the efficacy of the Lollisponge. The research design does not enhance acquisition and characterization of a strongly linked OSCC cancer and dissociation factor from non-cancer cells. Thus, not fulfilling the requirements for early cancer screening and control.
Human papilloma virus (HPV) can be an oncogenic contributor to induce verrucous carcinoma and oropharyngeal carcinoma. However, HPV oncogenic subtypes as shown in the study results are significantly unavailable to a majority of OSCC, which is driven to malignant pathogenesis severity by a variety of other factors. Although HPV probably supplies in some cancer cells a contributory loss of check-point control but is unlike to be a primary factor as the sites where HPV are in a reservoir differs from HPV driven cancers such as the base of the tongue, and oropharynx.
A lack of a link to driving factor for OSCC fulfilling one or both characteristics stated above at positive sensitive, specific, and accurate results is not present in this study. Instead based on the research design an indicator, HPV was chosen, and this factor is not a direct driver of OSCC.
For this reason, it is difficult to determine the significance of this study.
The study rigor and reproducibility are also questionable for these reasons.
Taken together, I would request modifications to attempt to comply with the state of the art and need requirements.
Author Response
Response to Reviewer 2’s Comments - Manuscript ID: cancers-3191603
Dear Reviewer 2,
Thank you for your thoughtful comments regarding our manuscript titled "Human Papilloma Virus (HPV) Detection in Oral Rinse vs Oral Sponge: a Preliminary Accuracy Report in Oral Cancer Patients" (Manuscript ID: cancers-3191603). We appreciate the opportunity to address these points and provide additional clarification.
Comment 1: Strengths:
The general premise for this study is an important aspect of cancer control which is the need for non-diagnostic methods to assess oral microenvironment using a non-invasive oral rinse with scope or Lollisponge. This approach facilitates development of rapid chairside detection and enhance communication between oral health professional and the patient, possible reducing time for early treatment if needed.
The presence of HPV within the oral cavity was used as an example of the efficacy of the device, Lollisponge. However, the choice of detection of HPV is a poor selection for this demonstration.
HPV as a reservoir can mediate oral mucosal immunity to reduce tumor immune surveillance but this is not addressed in this study.
This study in contrast simply used HPV subtype detection as an example of specificity, sensitivity and accuracy provided by the “Lollisponge.
- Response 1: Thank you for the time dedicated to reading our manuscript and for the highlighted strengths.
Comment 2: Weaknesses:
The primary question is: What is the value of the non-invasive technique Lolli sponge to detect important factors that characterize early malignant events linked to OSCC?
There are important requirement for any detection system, if it is to be more than just a mechanical approach of non-invasive technique. The brush non-invasive approach is not novel (e.g., oral cytology brushes), but has been employed in repeated studies for many years. Specifically, there is a US patent to use oral brushing using any device that provides a brushing action such as a Lollisponge, but importantly the patent describes use of oral brushing to obtain a miRNA classifier identifying OSCC compared to non-OSCC cells.
In brief malignant event drivers cause molecular-genomic alterations in the host oral keratinocyte-squamous cell (stem or stem cell like) that results in faulty genetic repairs and regulations with loss of check point cell cycle control and homeostatic maintenance. In addition, there is a depressive immunosuppression that occurs with loss of tumor immune surveillance. I mention these distinctive characteristics, as the state of the art for any cancer control or screening methodology as referenced above needs to comply with this need for assessing these characteristics.
Unfortunately, the approach described focuses on HPV detection which does not fulfill the requirements noted above and is therefore a poor choice to demonstrate the efficacy of the Lollisponge. The research design does not enhance acquisition and characterization of a strongly linked OSCC cancer and dissociation factor from non-cancer cells. Thus, not fulfilling the requirements for early cancer screening and control.
Human papilloma virus (HPV) can be an oncogenic contributor to induce verrucous carcinoma and oropharyngeal carcinoma. However, HPV oncogenic subtypes as shown in the study results are significantly unavailable to a majority of OSCC, which is driven to malignant pathogenesis severity by a variety of other factors. Although HPV probably supplies in some cancer cells a contributory loss of check-point control but is unlike to be a primary factor as the sites where HPV are in a reservoir differs from HPV driven cancers such as the base of the tongue, and oropharynx.
A lack of a link to driving factor for OSCC fulfilling one or both characteristics stated above at positive sensitive, specific, and accurate results is not present in this study. Instead based on the research design an indicator, HPV was chosen, and this factor is not a direct driver of OSCC.
For this reason, it is difficult to determine the significance of this study.
The study rigor and reproducibility are also questionable for these reasons.
Taken together, I would request modifications to attempt to comply with the state of the art and need requirements.
- Response 2: We thank you once again and agree with your comments regarding the minimally invasive brush approach for HPV oral sampling and the role of HPV in the molecular-genomic alterations of oral keratinocyte-squamous cells in OSCC development.
However, we would like to clarify certain points regarding the aim of our research. As stated in the manuscript, this pilot study aims to report the preliminary diagnostic accuracy of oral sampling using the LolliSponge, compared to a standard technique, the oral rinse, in identifying oral HPV-DNA in a cohort of patients with strictly oral SCC.
In response to the comment regarding the 'research design of the LolliSponge not fulfilling the requirements for assessing HPV's role in early cancer screening and control', we emphasize that this was not the objective of our pilot study. As previously mentioned, our goal was to compare the diagnostic accuracy of two non-invasive salivary sampling methods for HPV-DNA detection in strictly OSCC patients, with a primary focus on reporting the preliminary performance results of the oral sponge sampling (LolliSponge) compared to the standard technique of oral rinse in identifying oral HPV status. The study was not designed to assess the role of HPV in the oral carcinogenic process.
To our knowledge, this is the first study to analyze the capacity of the LolliSponge, and we compared it to the gold standard for HPV non-invasive salivary sampling (i.e., oral rinse). Our results show a high degree of diagnostic accuracy overlap between the LolliSponge and the Scope Oral Rinse. The LolliSponge could be a useful tool for identifying HPV infection and, when combined with targeted detection techniques (e.g., HPV methylation), could also help investigate the precise role of HPV in strictly OSCC. These are the future research directions of our team, and we hope to share them with the scientific community soon, after validating these preliminary findings in a larger cohort.
In conclusion, we thank the reviewer again for their time and valuable comments, and we hope to have clarified the concerns raised.
Best regards,
Fortunato Buttacavoli
Corresponding Author
DDS, PhD Student, Postgraduate Specialist in Pediatric Dentistry
Department of Precision Medicine in Medical, Surgical and Critical Care (Me.Pre.C.C.)
University of Palermo, Via Liborio Giuffrè, 5 - 90127
fortunato.buttacavoli@unipa.it

Reviewer 3 Report
Comments and Suggestions for Authors
Methodological Biases exist
Many Refs are missing
No trade marks in scientific articles
Limitations are missing
Extremely low study size
(The Authors must see my remarks)

Author Response
Response to Reviewer 3’s Comments - Manuscript ID: cancers-3191603
Dear Reviewer 3,
Thank you for your thoughtful comments and questions regarding our manuscript titled "Human Papilloma Virus (HPV) Detection in Oral Rinse vs Oral Sponge: a Preliminary Accuracy Report in Oral Cancer Patients" (Manuscript ID: cancers-3191603). We appreciate the opportunity to address these points and provide additional clarification.
Comment 1: Methodological Biases Exist
- Response 1:We acknowledge the concern regarding potential methodological biases. As this is a preliminary study, our approach was designed to provide initial insights into the use of non-invasive sampling methods for HPV detection. We understand that larger and more comprehensive studies will be necessary to confirm these findings and to address any potential biases. While our methodology was constrained by the study's scope, it aimed to lay a foundation for future research. Subsequent studies can build upon these preliminary results with more robust and diversified methodologies.
Comment 2: Many References Are Missing
- Response 2:We appreciate the feedback regarding the need for additional references. In our revision, we will carefully review and integrate relevant literature to address this issue. We recognize the importance of situating our findings within the broader body of research and will ensure that our references are comprehensive and up-to-date.
Comment 3: No Trade Marks in Scientific Articles
- Response 3:Thank you for highlighting the concern regarding the use of trademarks in scientific articles. We have revised the manuscript to remove all mentions of trademarks. However, we will defer to the editor’s decision on any further adjustments or specific formatting preferences to ensure the manuscript fully aligns with the journal's guidelines. We appreciate your attention to this detail, as it helps enhance the clarity and professionalism of our work.
Comment 4: Limitations Are Missing
- Response 4:We appreciate the suggestion to outline the study's limitations explicitly. In response, we have revised our discussion to include a detailed section addressing the small sample size, the limitations of non-invasive sampling methods, and potential biases (lines 644-654). This addition aims to provide a clearer understanding of the study's scope and underscores the need for further research.
Comment 5: Extremely Low Study Size
- Response 5:We acknowledge that the small sample size is a significant limitation of this study. As a preliminary investigation, our sample size was constrained by the scope and resources available. In the revised manuscript, we emphasize that the findings are intended to serve as a foundation for future research, which should involve larger cohorts to validate and expand upon these preliminary results.
Comment 6: The Authors Must See My Remarks
- Response 6:We appreciate your detailed feedback and have carefully reviewed and addressed your remarks throughout the text. Your insights have been invaluable in refining our study and enhancing its contribution to the field. We are committed to incorporating your suggestions and ensuring that the revised manuscript meets the highest standards of scientific rigor. For modifications such as changes to titles, subtitles, sections, or the removal of figures, we will defer to the final decision of the editor to ensure the optimal presentation of the work, in full accordance with the journal's guidelines.
We have thoroughly addressed all specific revisions requested directly within the text, providing appropriate responses and corrections directly on the file. Your thoughtful review and guidance have been instrumental in improving the manuscript, and we are grateful for your constructive input.
We thank you again for your constructive feedback and your positive engagement with our manuscript. Your insights have been invaluable in refining and improving the quality of our work.
Kind regards,
Fortunato Buttacavoli
Corresponding Author
DDS, PhD Student, Postgraduate Specialist in Pediatric Dentistry
Department of Precision Medicine in Medical, Surgical and Critical Care (Me.Pre.C.C.)
University of Palermo, Via Liborio Giuffrè, 5 - 90127
fortunato.buttacavoli@unipa.it

Round 2
Reviewer 1 Report
Comments and Suggestions for Authors
Dear Authors, Thank you for your revisions. We will do the usual checks prior to publication.
Reviewer 2 Report
Comments and Suggestions for Authors
THE PROBLEM WITH THIS STUDY IS THE RESEARCH DESIGN.
THE INVESTIGATION IS TO SHOW THE ACCURACY, SENSITIVITY AND SPECIFICTY OF THE LOLI-SPONGE DEVICE. THE REASON TO USE SUCH A DEVICE IS TO DETERMINE ITS CAPACITY FOR EARLY DETECTION ON CHARACTERISTICS OF ORAL SQUAMOUS CELL CARCINOMA. THE PROBLEM WHICH GOES UNCORRECTED FRON THE PREVIOUS VERISON OF THIS MANUSCRIPT IS THE CONCEPT THAT HPV, INCLUDING ONCOGENIC SUBTYPES ARE IMPORTANT DRIVERS OF ORAL CANCER ONCOGENESIS. THIS REATIONSHIP IS CLEARLY NOTED FOR OROPHARYNX CARCINOMA BUT UNFORTUNATELY THE IDENTICAL CASE CANNOT BE MADE FOR OSCC. THEREFORE THE USE OF HPV TO SHOW THE CAPACITY OF THE DEVICE TO LINK A FACTOR TO ORAL CANCER IS NOT MADE. WHAT THE STUDY RESULTS REVEALS IS THERE IS ANOTHER CYTOLOGIC TYPE DEVICE , OTHER THAN SEVERAL BRUSHES, THA CAN HARVEST BACTERIA, VIRUS, FUNGI, AND AN ARRAY OF GENOMIC MARKERS SUCH AS MIRNA, OTHER RNA TYPES AND DNA. THIS IS DOES NOT CONTRIBUTE WITH GREAT SIGNIFICANCE TO THE FIELD.